# Removal of Pollutants in Mine Wastewater by a Non-Cytotoxic Polymeric Bioflocculant from *Alcaligenes faecalis* HCB2

**DOI:** 10.3390/ijerph16204001

**Published:** 2019-10-19

**Authors:** Tsolanku Sidney Maliehe, Albertus Kotze Basson, Nkosinathi Goodman Dlamini

**Affiliations:** Department of Biochemistry and Microbiology, University of Zululand, KwaDlangezwa 3886, South Africa; BassonA@unizulu.ac.za

**Keywords:** bioflocculant, flocculating activity, cytotoxicity, removal efficiency

## Abstract

Bioflocculation is a physicochemical technique often employed to efficiently remove colloidal water pollutants. Consequently, in this study, a bioflocculant was produced, characterised and applied to remove pollutants in mine wastewater. The maximum flocculation activity of 92% was recorded at 30 °C, pH 9.0 when maltose and urea were used as energy sources and 72 h of fermentation at the inoculum size of 1% (*v*/*v*). K^+^ proved to be a favourable cation. The bioflocculant yield of 4 g/L was obtained. Scanning electron microscopy illustrated a hexagonal-like structure of the bioflocculant. It is composed of carbohydrates and proteins in mass proportion of 88.6 and 9.5%, respectively. The Fourier transform infrared spectrum revealed the presence of hydroxyl, amide and amino functional groups. More than 73% of the bioflocculant was obtained after exposure to 600 °C using the thermogravimetric analyser. Human embryonic kidney 293 (HEK 293) cells exhibited 95% viability after being treated with 200 µg/µL of the bioflocculant. The flocculation mechanisms were proposed to be as a result of a double layer compression by K^+^, chemical reactions and bridging mechanism. The removal efficiencies of 59, 72, and 75% on biological oxygen demand, chemical oxygen demand and sulphur, were obtained respectively. Thus, the bioflocculant have potential use in wastewater treatment.

## 1. Introduction

In developing countries, approximately 90% of wastewater is discharged untreated into water bodies [1]. This threatens aquatic life and contributes to water incompatibility as palatable and potable molecule [2]. Colloids are major water pollutants. They are often heterogeneous matter characterized by thermodynamical instability and kinetical non-labile. Colloids tend not to settle in solution under gravity [3]. The electrokinetic properties on their surfaces make their removal a great challenge [4]. Flocculation is a physicochemical technique often employed to destabilise colloids and enhance their agglomeration to form large flocs that can easily settle-out in solution and removed by sedimentation, filtration or flotation [5]. Inorganic flocculants such as hydrolysable salts of iron and aluminium and synthetic flocculants such as polyacrylamide have been predominantly and extensively utilized in wastewater treatment in the past decades [6]. However, these flocculants have been reported to be toxic to humans and are environmentally unfriendly [7]. Their monomeric units are linked to causing neurotoxic and carcinogenic effects on humans. Moreover, they possess un-hydrolysable bonds, which render them undegradable [8]. Therefore, due to the drawbacks of the inorganic and synthetic flocculants, biological wastewater treatment agents such as bioflocculants, have recently gained attention [9]. Bioflocculants are macromolecular polymers of carbohydrates, proteins, lipids or nucleic acids secreted by microorganisms during metabolic processes [10]. Due to their eco-friendliness and innocuousness to humans, they are viewed as alternatives to chemical and synthetic flocculants [11,12]. They have been effectively used in flocculating inorganic solid suspensions [13], microalgae [14] and heavy metals [15]. Nevertheless, low bioflocculant yields have been the limiting factor for their industrial applications. Although there are many microorganisms investigated, there is still a large number of new organisms, especially from the untapped regions of the oceans, which may have high bioflocculant production capabilities and unique bioflocculant features [16]. Therefore, the search for new bioflocculant-producers that can display high bioflocculant yields and flocculating activities, is attracting much research attention.

In this study, we optimised the culture conditions of a novel bioflocculant-producing strain, previously identified as *Alcaligenes faecalis* HCB2 by physiological characteristics and 16S rRNA sequence analysis. The bioflocculant was characterized and we hypothetically evaluated its flocculation mechanism in kaolin solution. Its ability to remove organic matter measured as biological oxygen demand, chemical oxygen demand and sulphur from mine wastewater was determined in comparison to chemical flocculants.

## 2. Materials and Methods

### 2.1. Chemicals and Production Medium

All chemicals, reagents and media used were procured from Sigma-Aldrich (St Louis, MO, USA). The standard production medium by Zhang et al. [13], was used. The medium composed of glucose (20.0 g), KH_2_PO_4_ (2.0 g), K_2_HPO_4_ (5.0 g), (NH_4_)_2_SO_4_ (0.2 g), NaCl (0.1 g), CH_4_N_2_O (0.5 g), MgSO_4_ (0.2 g), and yeast extract (0.5 g) in a litre of the filtered seawater. The media were autoclaved at 121 °C for 15 min.

### 2.2. Source of Bacterium

*Alcaligenes faecalis* HCB2 was previously isolated from the sediment sample from Sodwana Bay in the Province of KwaZulu-Natal in South Africa (28°45′ S 31°54′ E). It has then been stored at 20% glycerol at −80 °C in the Department of Biochemistry and Microbiology at the University of Zululand, South Africa. Prior to being used, *A. faecalis* HCB2 was resuscitated on nutrient agar (NA).

### 2.3. Determination of Bioflocculant Production

A loopful of *A. faecalis* HCB2 was inoculated into a 250 mL flask containing 50 mL of sterilised production medium. The mixture was then incubated at 30 °C in a shaker at 165 rpm for 72 h. The culture broth was centrifuged at 8000 rpm for 30 min at 4 °C to remove cells and the flocculating activity was determined spectrophotometrically at an optical density of 550 nm according to [17]. Prior to the determination of the flocculating activity, 100 mL of kaolin suspension (4 g/L) was measured into a 250 mL flask. Three millilitres of 1% *w*/*v* CaCl_2_ and 2 mL of the obtained cell-free supernatant were added. The mixture was shaken for 1 min and then poured into 100 mL measuring cylinder. A control test was done by adding the same volume of kaolin suspension with 3 mL of 1% *w*/*v* CaCl_2_ and 2 mL of autoclaved distilled water. Measuring cylinders were then allowed to stand for 5 min at room temperature. The formula used for the determination of the flocculating activity was as follows:
Flocculating activity (%) = [(A−B)/A]×100
where A is the optical density of a control at 550 nm and B is the optical density of a sample at 550 nm.

### 2.4. Optimisation of Culture Conditions

The optimum medium culture conditions for bioflocculant production were assessed. The inoculum sizes were varied in a range of 0.5–2.5 mL, (*v*/*v*), representing (1–5%) and the flocculating activity was determined, thereafter [18]. The 20 g/L of glucose used in the original medium was replaced with various carbon sources (sucrose, maltose, starch, fructose, lactose, and molasses) for determination of their effect on flocculating activity. Various nitrogen sources (peptone, urea, yeast, ammonia, and casein) of 1.2 g/L, were utilized individually in the medium and flocculating activity determined [9]. The mixing speeds in a range of 0–220 rpm, pH of the culture medium (3–12), cultivation temperatures (20–50 °C) were also varied, respectively. Cations such as K^+^, Na^+^, Li^+^, Ca^2+^, Mn^2+^, Ba^2+^, and Fe^3+^, in volumes of 3 ml of 1% *w*/*v*, were evaluated in the place of CaCl_2_ [19]. For the effect of time course on the flocculating activity, the bacterium was cultured under the previously obtained optimal growth conditions. Samples were drawn every 12 h over a period of 108 h and flocculating activity, optical density (OD_550 nm_) and pH were determined [20].

### 2.5. Extraction and Purification of the Bioflocculant

The extraction and purification of the bioflocculant from the bioflocculant-rich broth were done in accordance with the methods of Choi et al. [21]. The culture broth was centrifuged at 8000 rpm at 4 °C for 30 min to remove bacterial cells. Two volumes of ethanol were added to the supernatant. The solution was thoroughly agitated and left at 4 °C for 12 h. The precipitate was freeze-dried to obtain crude bioflocculant and the crude product was dissolved in distilled water to yield a solution (1% *w*/*v*). A volume of a mixture of chloroform and methanol (2:1 *v*/*v*) was added, agitated and left at room temperature for 12 h. The supernatant was then centrifuged (8000 rpm for 30 min, 4 °C) and freeze-dried in order to obtain a purified bioflocculant.

### 2.6. Effect of Bioflocculant Concentration and PH on Flocculating Activity

The influence of bioflocculant concentration towards the flocculating activity was assayed by varying different concentrations of the obtained, purified bioflocculant (0.2–1 mg/mL). The flocculating activities were determined as described previously. The effect of pH on flocculating activity was investigated. Briefly, the pH of kaolin suspension (100 mL, 4 g/L) was adjusted in a range of pH 3–12 using 0.1 M HCl and 0.1 M NaOH. The flocculating activity was determined using the jar test procedure as described previously [22].

### 2.7. Physical and Chemical Characteristics

The surface of structures of the bioflocculant, kaolin particles and floc were examined using a scanning electron microscope (SEM-Sipma-VP-03-67) [23]. Total sugar and protein contents were determined by phenol-sulphuric acid method [24,25]. The elemental analysis was carried out with scanning electron microscope-energy dispersive x-ray detector (SEM-EDX) (Oxford Instruments-X-Max ^N^) [26]. Prior to SEM analysis, 5 mg of the bioflocculant was placed on a slide coated with silicon and fixed by spin coater (1000 rpm, 1 min). The functional groups of the bioflocculant were determined by Fourier transform infrared (IR) spectroscopy [27]. Its degradation temperature was studied using a thermo-gravimetric instrument (Perkin Elmen Pyris 6 TGA). The bioflocculant was heated from 22 to 600 °C at a rate constant of 10 °C min^−1^ under constant flow of nitrogen gas [28].

### 2.8. Cytotoxicity Assay of Bioflocculant

The cytotoxicity was measured according to the method by Mosman. [29]. Human embryonic kidney 293 (HEK 293) cells were grown to confluency in 25 cm^3^ flasks. The cells were trypsinised and plated into 48 well plates. Cells were incubated overnight at 37 °C. The old medium was supplemented with the fresh medium (MEM + Glutmax + antibiotics). Bioflocculant was added in triplicate and incubated for 4 h. Thereafter, the medium was removed and replaced by complete medium (MEM + Glutmax + antibiotics +10% Fetal bovine serum). After 48 h, the cells were subjected to 200 μL of 3-(4,5-dimethylthiazol-2-yl)-2,5-diphenyl tetrazolium bromide) (MTT) of the concentration of 5 mg/mL in phosphate-buffered saline (PBS) and 200 μL medium was added to each well and incubated at 37 °C for 4 h. Thereafter, the medium was aspirated from the wells and the formed formazan crystals were solubilized in 200 μL of dimethyl sulfoxide (DMSO). The optical density of the solutions was read at 570 nm using a microplate reader. The effect of the bioflocculant on HEK 293 cell viability was measured as the percentage of tetrazolium salt reduction by viable HEK 293 cells against the untreated cells. The formula used for the determination of the cell viability was as follows: Cell viability (%) = (OD_untreated cells_ − OD_treated cells_) / OD_untreated cells_ × 100.

### 2.9. Proposed Flocculation Mechanism

The flocculation mechanism of the bioflocculant was proposed after the zeta potential was measured by Zetasizer Nano (Malvern, UK). The zeta potentials of the bioflocculant solution (0.2 mg/mL), kaolin clay suspension (4 g/L), 100 ml of kaolin solution in the presence of 3 mL of 1% (*v*/*w*) of KCl and kaolin flocculated by the bioflocculant (2 mL) in the presence of 3 mL of 1% (*v*/*w*) of KCl were investigated at pH 7 at 25 °C [30].

### 2.10. Removal Efficiency of Bioflocculant on Wastewater

Wastewater sample was collected aseptically from Tendele Coal Mine wastewater plant for evaluation of the removal efficiency (RE) of pollutants by the bioflocculant. pH was adjusted to 7 using 0.1 M HCl and 0.1 M NaOH. The Jar test was used in accordance with the method by Okaiyeto et al. [6]. Briefly, 3 mL of 1% (*w*/*v*) KCl and 2 ml of the bioflocculant solution were poured into 100 mL of wastewater sample. The mixture was shaken at 200 rpm for 3 min. The speed was reduced to 40 rpm and then allowed to shake for 5 min at room temperature. The supernatant was collected. Thereafter, chemical oxygen demand (COD), biological oxygen demand (BOD) and sulphur (S) of the untreated and treated (supernatant) samples were measured using test kits in accordance to the manufacturer’s guidelines. The cells were read at 680 nm using spectrophotometer (Spectro-quant, Merck Pharo 100). Removal efficiencies on BOD, COD and S were measured as follows:
Removal efficiency (%) = (I_o_ − S/ I_o_) × 100
where I_o_ and S are the initial and final values obtained before and after treatment with bioflocculant. The removal efficiency of the bioflocculant was compared to those of Alum and ferric chloride, respectively.

### 2.11. Software and Statistical Analysis

All the experimentations were done in triplicates and the data were subjected to one-way analysis of variance using Graph Pad Prism™ 6.1. Arrow bars represented the standard deviation and values with different alphabets represent the significant difference (*p* < 0.005).

## 3. Results and Discussion

Although, there are many studies conducted about bioflocculants, flocculating activity and low yields are still the major limiting factors with regard to their biotechnological applications [13]. Consequently, it is important to identify novel bioflocculant-producers especially from unusual environments such as sea, with high bioflocculant production and improved flocculating efficiencies. Thus, in this study, a previously identified bioflocculant producing strain and its bioflocculant were studied.

### 3.1. Effect of Inoculum Size

Inoculum size is one of the important factors that affect bioflocculant production [31]. *A. faecalis* HCB2 achieved the highest flocculating activity of 71% when an inoculum size of 1% (*v*/*v*) was used (Table 1). The flocculating activity decreased when an inoculum size was further increased. A large inoculum size tends to cause a niche of the bioflocculant-producers to overlap excessively, thereby suppressing bioflocculant production and thus, flocculating activity [32]. This phenomen was observed in this study, whereby the lowest inoculum size yielded high flocculating activity in comparison to others. The results were in agreement with those of Agunbiade et al. [33], whereby the inoculum size of 1% (*v*/*v*) of *Streptomyces platensis* FJ 486385.1 was preferred. However, the results are slightly different to those of *Brachybacterium*, whereby the highest flocculating activity was observed when inoculum size of 2% (*v*/*v*) was used [34]. Therefore, the optimum inoculum size of 1% (*v*/*v*) was used throughout the research work.

### 3.2. Effect of Carbon and Nitrogen Sources

Carbon and nitrogen sources are needed by microorganisms for energy required for all biosynthesis processes leading to the production of bioflocculants and cell maintenance [6]. Several sugars were investigated for determination of their effect on flocculating activity (Table 1). Compared to other sugars, maltose showed an outstanding flocculating activity of 91%. Similarly, Abd-El-Haleem et al. [35], found the same results with maltose being the preferred carbon source during bioflocculant production by *Bacillus aryabhattai* strain PSK1. Although molasses was the most economical carbon source used, it gave low flocculating activity (38%). However, almost all tested carbon sources, with the exception of molasses, can be effectively utilized by *A. faecalis* HCB2 as they all demonstrated flocculating activities above 60%. Urea was the nitrogen source of choice since it demonstrated the highest activity (97%) (Table 1). Yeast extract showed the lowest flocculating activity (22%). Kurane and Nohata. [36], reported that *Alcaligenes latus* had maximum flocculating activity when urea and yeast extract were used as nitrogen sources. Cosa et al. [20], stated that organic nitrogen sources are the best energy sources since they are easily absorbed by the cells and enhance cell activities of bioflocculant-producers.

### 3.3. Effect of Metal Cations

Cations enhance the flocculation rate by neutralising and stabilising the residual negative net surface charge of the bioflocculant functional groups [37]. The effect of various metal ions on the flocculating activity of the bioflocculant produced by *A. faecalis* HCB2 is shown in Table 1. The highest flocculating activity of 78% was observed when monovalent cation (K^+^) was used. Virtually, both monovalent and divalent cations showed better flocculating activities than the trivalent cations. These findings contrast the work of Wu and Ye [38], who reported that monovalent cations reduce the strength of the bonds and result in a loose structure of flocs, thus decreasing the flocculating activity.

### 3.4. Effect of Shaking Speed

Different bacterial strains require different shaking speeds to effectively produce high yields of bioflocculants [39]. Figure 1 illustrated the effect of mixing speed on flocculating activity. The flocculating activity increased gradually in proportion to the increase in the agitation speed and reached its maximum (77.9%) at 165 rpm. This speed enhanced the optimum concentration of oxygen to be dissolved and allowed the even distribution of nutrients and proper mixing of growing cells of *A. faecalis* HCB2 [40,41]. Above 165 rpm, the flocculation activity decreased slightly but insignificantly. Higher agitation speed may have resulted in high shear stress and impacted negatively on cell growth and bioflocculant production [42]. The results were almost similar to those of [43], whereby the mixing speed of 160 rpm was optimum and produced the highest flocculating activity (92.2%) of the produced bioflocculant.

### 3.5. Effect of Cultivation Temperature

The catalytic activity of enzymes responsible for bioflocculant production depends on the specific optimum temperature [44]. Thus, the effect of temperature on flocculating activity was assessed and the results presented in Figure 2. When the temperature was 30 °C, the flocculating activity reached a maximum of 86%. The temperatures above or below 30 °C led to a slight decrease in the flocculating activities. The temperature above 30 °C might have influenced the loss of the catalytic configuration of the enzymes, leading to the loss of enzymatic activities in metabolic reactions involved in bioflocculant production [45]. However, lower temperatures might have propelled *A. faecalis* HCB2 to hibernate and cease to produce sufficient bioflocculant [46].

### 3.6. Effect of Initial PH on Flocculating Activity

The initial pH of the culture medium determines the electrification of the cells and oxidation-reduction potential which could influence the absorption of nutrients in the production medium and enzymatic reactions [11]. It was observed that the lowest flocculating activity (53%) was obtained at pH 3 (Figure 3). The bioflocculant was produced mostly at the alkaline conditions, where the highest flocculating activity of 85 % was observed at pH 9. Similar results were observed by Ugbenyen et al. [47], whereby *Bacillus* sp. Gilbert had the highest flocculating activity of 93.77% at pH 9. On the contrary, Yim et al. [48] observed optimum flocculating activity at pH 4 for *Gyrodinium impudicum*. This suggests that the bioflocculant-producers prefer different initial pH.

### 3.7. Effect of Time on Flocculating Activity, Cell Number and PH

The bacterium growth curve in relation to the flocculating activity, cell number (OD_550_), and pH is illustrated in Figure 4. The flocculating activity was at a peak of 90% at the stationary phase (72 h). After 72 h, it declined significantly. The decline might have been due to cell autolysis or the presence of degrading enzymes [49]. As expected, the OD of the media increased throughout the cultivation time. Between 48 to 84 h, the OD increased constantly. This might be owed to the exponential growth of the bacterium due to favourable culture conditions. After 84 h, there was a slight increase in OD of the medium. The increase might have been due to the fact that, as the nutrients got depleted, the accumulation of dead cells and the waste products of metabolic activities also increased [50]. Nevertheless, the observed increase in optical density and flocculating activity within 72 h, suggested that the bioflocculant was produced by biosynthesis and not cell autolysis [8]. The pH of the fermentation medium constantly decreased in the first 84 h of cultivation. The decrease might have been due to the organic acidic excreted by *A. faecalis* HCB2 during metabolic reactions [27]. However, after 84 h, it increased insignificantly. The buffering activity assumed to be from the alkaline components excreted by the bacterium.

### 3.8. Bioflocculant Yield

Yield and productivity are good quantitative analysis that indicates how bacterial cells convert components in culture medium into effective bioflocculants. *A. faecalis* HCB2 yielded about 4 g/L of the bioflocculant. The high yield might be owed to the ability of the strain to optimally survive and produce the bioflocculant in the optimised culture conditions and the polarity of the solvents used during extraction. Bioflocculant was much higher than most of bioflocculants produced by pure bacterial strains which are often less than 2 g/L [49].

### 3.9. Bioflocculant Dosage Size

An inadequate bioflocculant concentration does not sufficiently neutralize some of the negative charges on colloidal particles and thus results in low activity [51]. On the other hand, excess dosage sizes negatively affect the settling and stability of the flocs due to the increased viscosity [52]. As illustrated in Table 2, the concentration of 0.8 mg/ml gave the highest flocculating activity of 85.6%. The concentration between 0.2–0.6 mg/mL may have not permitted the bridging phenomena to happen effectively hence it had low flocculating activities. Moreover, the concentration above 0.8 mg/mL may have also caused competition and repulsion of negatively charged kaolin particles, consequently blocking the binding sites available on the surfaces of kaolin particles for the formation of interparticle bridges and leading to re-stabilization of the kaolin particles in solution, resulting in a decreased flocculating activity [7].

### 3.10. Effect of PH on Flocculating Activity of the Purified Bioflocculant

The pH of reaction mixtures is a key factor that influences the flocculation process [53]. pH may alter bioflocculant charge status and the surface characteristics of colloidal particles in suspension and consequently affect flocculating activities [51] Figure 5 presents the effect of pH on the flocculating activity of the purified bioflocculant. The bioflocculant was effective within a wide range of pH (5–12), giving flocculating activities equal to or above 85%. The highest flocculating activity of 93% was observed at pH 7.0 and the least (69%), occurred at pH 3. The slight decrease in flocculating activities in strong acidic conditions pH (3–4) might have been due to the protein denaturation in the bioflocculant. Therefore, pH 7 was used as the optimum pH in the successive experiments.

### 3.11. Surface Morphological Structure

The surface morphological structure of bioflocculants plays a vital role in the flocculation process [13]. It is accountable for effective or poor flocculating activities of bioflocculants. Scanning electron microscope images are shown in Figure 6. SEM image demonstrated the hexagonal-like shape of the bioflocculant. Its configuration might be accountable for its high flocculating activity. The floc appeared clustered. It appeared clustered due to flocculation process, which enabled kaolin particles to be adsorbed onto the binding sites of the bioflocculant, and thus, larger flocs were formed because of this interaction. Kaolin particles appeared to be fine and smooth in structure.

### 3.12. Bioflocculant Composition

Bioflocculants compose of polysaccharides, lipids, proteins, nucleic acids, and other polymeric compounds [53]. The bioflocculant (1 mg/mL) revealed to have high content of total carbohydrates (0.886 mg/mL) and low total protein content (0.095 mg/mL) (Table 3). The results were similar to those of Toeda and Kurane. [54]. Whereby the bioflocculant from *Alcaligenes cupidus* KT201 was mainly composed of carbohydrates. The composition of both carbohydrates and proteins suggested that the bioflocculant has multiple functional groups. Multiple functional groups are indicative for many adsorption sites for the colloidal particles and this can lead to high flocculating efficiency [55]. Carbohydrates derivatives in bioflocculant were, therefore, presumed to be the most responsible components for the experientially high flocculating activity as the bioflocculant is predominantly carbohydrates.

### 3.13. Functional Groups of the Bioflocculant

The binding capability of the bioflocculants depends on the number of functional groups in their molecular chains [56]. The analysed functional groups of the bioflocculant are shown in Figure 7. The IR displayed a broad stretching peak at 3368 cm^−1^, indicative of the presence of the hydroxyl and amino groups. The peak at 1641 cm^−1^, was a characteristic of C=O, which suggested an amide group and the absorptive peak at 1019 cm^−1^ with a weak bond between C–N was observed, presentative of the amino group. The small weak absorptive peak at 530 cm^−1^, showed characteristic of halo compound CBr. In summary, the IR spectrum revealed the presence of hydroxyl, carboxyl (in the amide group) and amino groups in the produced bioflocculant. The observed functional groups might have contributed to the flocculating activity and stability of the bioflocculant. Moreover, the presence of the carboxyl groups on the molecular chain of bioflocculant might have enabled the chain to spread out as a result of electrostatic repulsion and the stretched chains provided more effective sites for attachment of colloidal kaolin particles in suspension [57]. The obtained results were comparable to those of other studies [58,59].

### 3.14. Elemental Composition of the Bioflocculant

The elemental composition of the bioflocculants plays a vital role in bioflocculant structure and flocculating activity [20]. Various elements bring about flexibility and stability of the bioflocculants. Quantitative analysis of the elemental composition of the bioflocculant was assessed and results are shown in Figure 8. The elemental spectrum shows the absorption peaks indicative of O:C:P:Ca:Mg:S:Na:N:Cl accounting for 62.1:18.7:8.1:6.6:3.0:0.6:0.3:0.3:0.3 (%, *w*/*w*), respectively. The presence of carbon, oxygen and nitrogen elements confirms the bioflocculant as a glycoprotein polymer [60]. The results were almost similar to those of the bioflocculant from marine *Halobacillus* which had multiple elements [20].

### 3.15. Thermogravimetric Analysis of the Bioflocculant

The thermogravimetric analysis was used to establish the pyrolysis profile of the purified bioflocculant (Figure 9). There was an initial weight loss of 9.31% observed between 35 and 151.65 °C. The initial weight loss was mainly due to the loss of the moisture content. According to Kumar and Anand. [61], the moisture content in bioflocculant is due to the presence of hydroxyl and carboxyl groups. High carboxyl content tends to promote higher affinity of the carbohydrates for water molecules. A further increase in temperature resulted in decomposition of the bioflocculant at 210, 370 and 480 °C. More than 73% of the bioflocculant was obtained after exposure to the highest temperature of 600 °C. Summarily, in accordance with the observations, it was deduced that the bioflocculant was thermostable. Moreover, when compared to other bioflocculants, the test bioflocculant had better thermal stability [33].

### 3.16. Cytotoxic Effect of the Bioflocculant on HEK 293 Cell-Line

Although bioflocculants have been generally affirmed as non-toxic compounds, for biosafety reasons, they still need to be tested for their toxicity before use [60]. This is due to the fact that some may impose toxic effects [5]. In this study, the bioflocculant showed a margin of safety, as it had no significant cytotoxic effect on the tested cell line (Figure 10). HEK 293 cell-line exhibited 95% viability after being treated with the highest concentration of bioflocculant (200 µg/µL). The results affirmed the probable safe utilization of the bioflocculants in different applications. The results were in agreement with those of Sharma et al. [62], where the exopolymer produced by *Acinetobacter haemolyticus* showed no toxicity on sheep blood cells. The same biopolymer had no effect in an in-vivo study done on rats, where no clinical symptoms were observed. El-Rouby et al. [63], also found similar results whereby bioflocculants from bacterial strains showed no clinical toxic effects on rats.

### 3.17. Proposed Flocculating Mechanisms of the Bioflocculant

Bioflocculants cause flocculation of particles by two mechanisms namely: Charge neutralization and bridging [64]. Charge neutralization occurs when the particle and bioflocculants are charged oppositely, while bridging mechanism occurs when the segments of the bioflocculant chains or functional groups are absorbed onto the colloids, thus binding the colloidal particles together [30]. The electric charges of bioflocculant and kaolin particles are both negative (Table 4). Addition of K^+^ to kaolin suspension and kaolin suspension plus bioflocculant resulted in the reduction of zeta-potential. When the negative charge is reduced or totally abolished, the repulsion force becomes terminated and particles easily agglomerate [2]. Thus, K^+^ increased the adsorption of the bioflocculant on the surface of colloidal kaolin particles by decreasing the negative charge on bioflocculant and kaolin particles. Thus, permitting the bioflocculant and kaolin particles in solution to draw nearer to each other and chemically bond K^+^ compressed the double layer of colloidal kaolin particles, weakened the static repulsive force and enhanced the bioflocculant to form aggregates with colloidal kaolin particles in solution. Summarily, K^+^ aided bridging flocculation between bioflocculant and kaolin particles. In addition, the presence of hydroxyl and carboxyl groups of the bioflocculant suggested that the chemical interactions might have included the formation of the ionic and hydrogen bonds [47].

### 3.18. Proposed Flocculating Mechanism of the Bioflocculant

The ability of the bioflocculant to reduce suspended particles in wastewater is shown in Table 5. When compared to the conventional chemical flocculants, bioflocculant showed better removal activities on all tested parameters. Generally, the removal efficiencies of these bioflocculants were attributed to its surface structure, chemical components and functional groups. The effectiveness illustrated by the bioflocculant implied that it has potential use as an alternative means to the currently predominant conventional flocculants. The results were almost similar to those of Zhang et al. [13], whereby bioflocculant MMF1 had the maximum removal efficiency of 79.2% on COD.

## 4. Conclusions

The study showed that *A. faecalis* HCB2 produced high yield and flocculating activity when glucose and inorganic ammonium sulphate were used in optimum environmental conditions. The characterization results proved the bioflocculant as a heat-stable glycoprotein molecule. It had hydroxyl, amide and amino groups as its main functional groups. The cation mediated charge and bridging mechanisms between bioflocculant and kaolin particles. The bioflocculant had an insignificant cytotoxic effect and good removal efficiencies on BOD, COD, and S on coal mining wastewater. The revealed properties suggested its potential in industrial applications. For further studies, the bioflocculant will be applied on dye removal.

## Figures and Tables

**Figure 1 ijerph-16-04001-f001:**
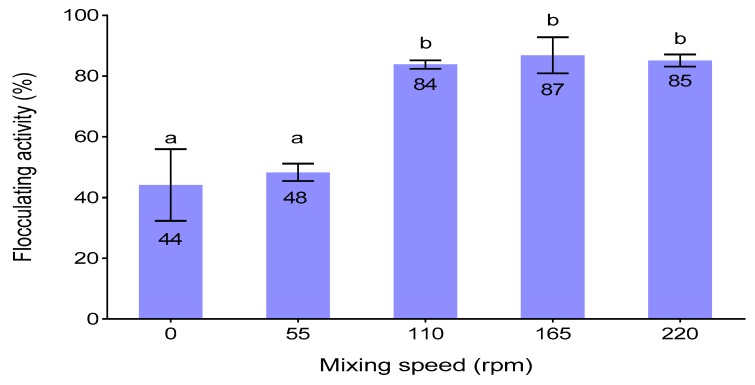
Effect of shaking speed on flocculating activity different letters (a & b) denotes statistical significance at (*p* < 0.05).

**Figure 2 ijerph-16-04001-f002:**
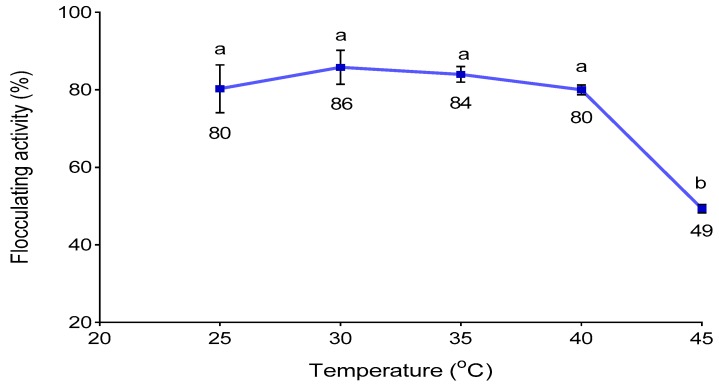
Effect of fermentation temperature on flocculating activity different letters (a & b) denotes statistical significance at (*p* < 0.05).

**Figure 3 ijerph-16-04001-f003:**
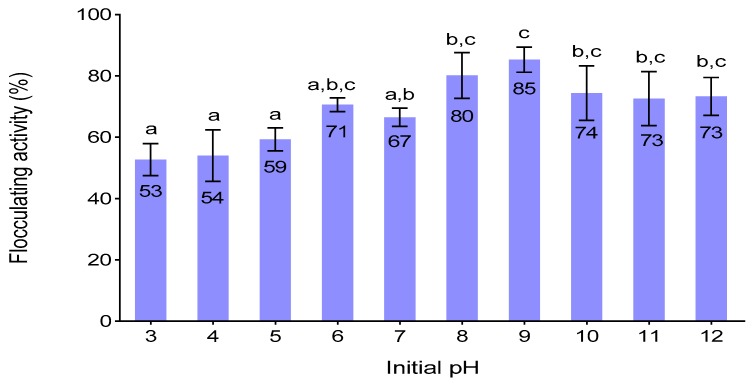
Effect of initial pH on flocculating activity different letters (a, b & c) denotes statistical significance at (*p* < 0.05).

**Figure 4 ijerph-16-04001-f004:**
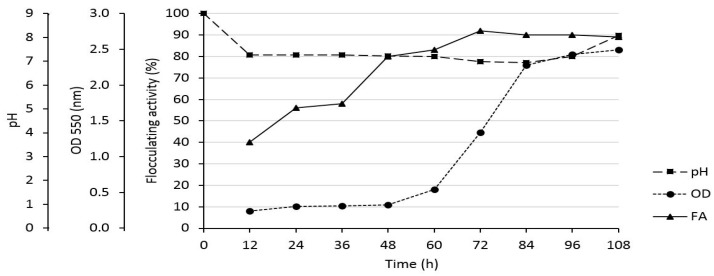
The effect of time on FA, initial pH and OD.

**Figure 5 ijerph-16-04001-f005:**
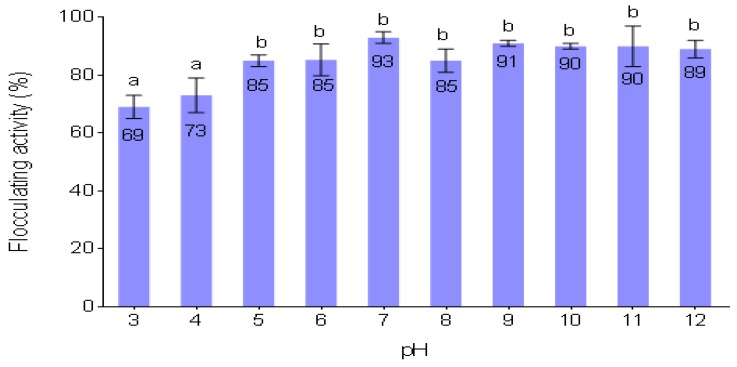
Effect of pH on the purified bioflocculant different letters (a & b) denotes statistical significance at (*p* < 0.05).

**Figure 6 ijerph-16-04001-f006:**
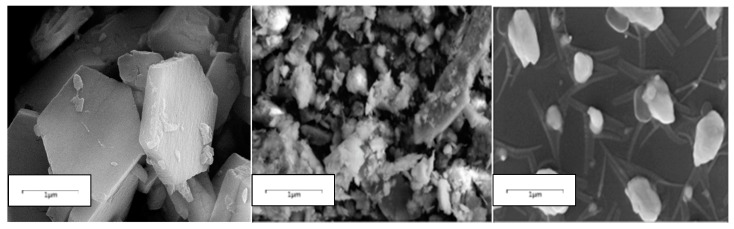
SEM analysis. SEM surface images of the bioflocculant (**left**), flocculated kaolin particles (**middle**) and kaolin particles (**right**).

**Figure 7 ijerph-16-04001-f007:**
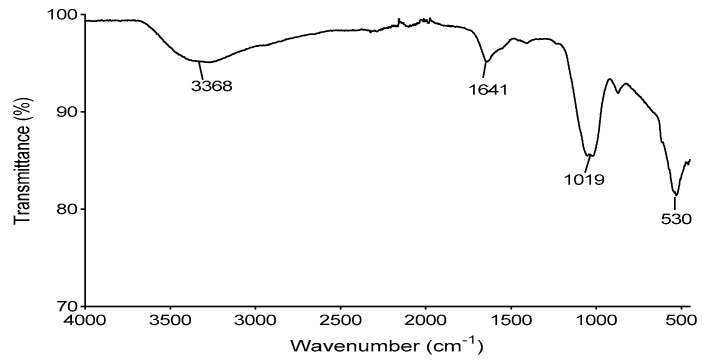
IR spectrophotometry analysis of bioflocculant.

**Figure 8 ijerph-16-04001-f008:**
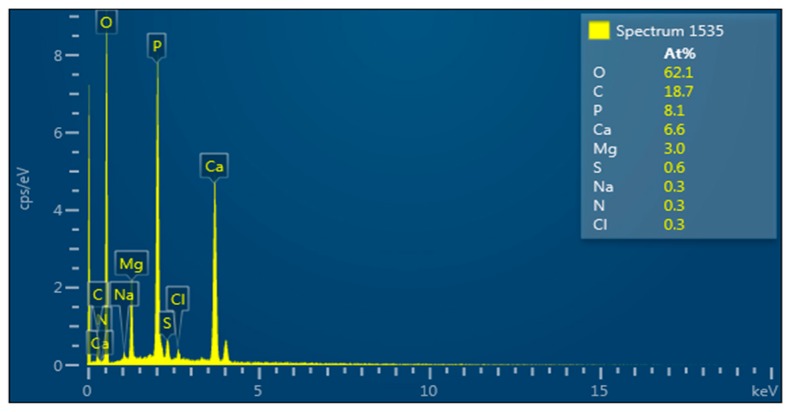
Elemental analysis of bioflocculant.

**Figure 9 ijerph-16-04001-f009:**
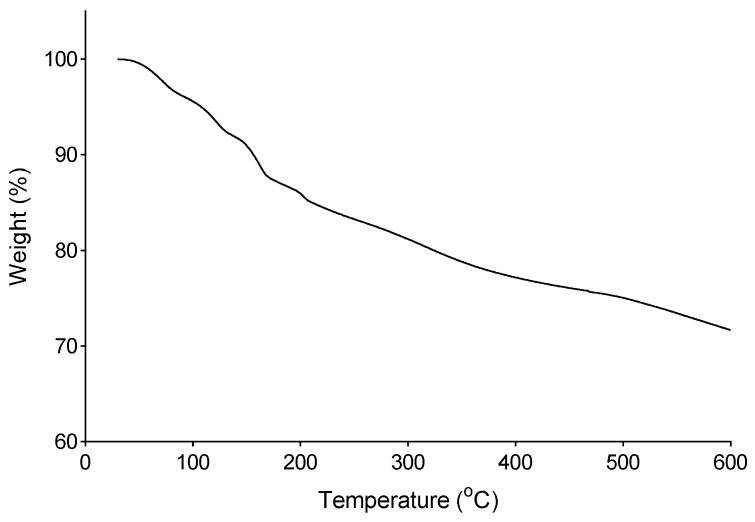
Thermo-gravimetric analysis of the bioflocculant.

**Figure 10 ijerph-16-04001-f010:**
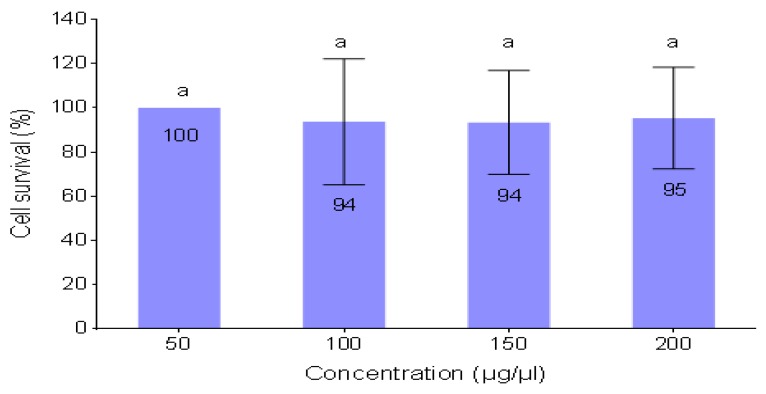
In-vitro cytotoxicity of different concentrations of bioflocculant on HEK293 different letters (a) denotes statistical significance at (*p* < 0.05).

**Table 1 ijerph-16-04001-t001:** Effect of inoculum size, carbon sources, nitrogen sources and cations on flocculation.

Inoculum Size (%)	FA (%) ± SD	Carbon Source	FA (%) ± SD	Nitrogen Source	FA (%) ± SD	Cations	FA (%) ± SD
1	70.8 ± 5.50 ^a^	Molasses	38.6 ± 23.20 ^c^	Yeast extract	22.4 ± 7.05 ^c^	Control	49.5 ± 3.35 ^a,c^
2	68.5 ± 3.46 ^a^	Sucrose	61.4 ± 0.78 ^b,c^	Casein	34.0 ± 4.09 ^c^	Na^+^	62.3 ± 7.28 ^a,b^
3	68.3 ± 3.52 ^a^	Glucose	70.7 ± 3.55 ^a,b^	Peptone	80.4 ± 1.25 ^b^	Li^+^	75.4 ± 2.31 ^b^
4	67.4 ± 6.78 ^a^	Starch	79.1 ± 2.81 ^a,b^	(NH_4_)_2_SO_4_	89.2 ± 6.88 ^a,b^	K^+^	78.1 ± 2.52 ^b^
5	67.7 ± 7.46 ^a^	Lactose	81.2 ± 1.80 ^a,b^	Urea	97.4 ± 0.84 ^a^	Mn^2+^	63.2 ± 6.78 ^a,b^
		Fructose	88.1 ± 1.85 ^a^			Ba^2+^	63.9 ± 2.08 ^a,b^
		Maltose	90.6 ± 2.11 ^a^			Ca^2+^	71.2 ± 5.42 ^b^
						Fe^3+^	31.1 ± 3.15 ^d^

FA denotes flocculating activity while SD denotes standard deviation. Different letters (a, b & c) denotes statistical significance at (*p* < 0.05).

**Table 2 ijerph-16-04001-t002:** Indicated the effect of different bioflocculant concentrations on the flocculating activity of the purified bioflocculant.

Dosage (mg/mL)	FA (%) ± SD
0.2	80.4 ± 1.04 ^a^
0.4	78.7 ± 1.29 ^a^
0.6	78.5 ± 2.72 ^a^
0.8	85.6 ± 1.35 ^b^
1	84.8 ± 1.04 ^b^

FA denotes flocculating activity while SD denotes standard deviation. Different letters (a & b) denotes statistical significance at (*p* < 0.05).

**Table 3 ijerph-16-04001-t003:** Chemical components of the purified bioflocculant.

Samples	Concentration (mg/mL)
Carbohydrates	0.886
Proteins	0.095

**Table 4 ijerph-16-04001-t004:** Zeta potential of different samples.

Samples	Zeta Potential (mV)
Bioflocculant	−17.1 ± 0.7
Kaolin particles	−6.59 ± 3.0
Kaolin particles with K^+^	−7.01 ± 1.0
Kaolin particles flocculated with bioflocculant in the presence of K^+^	−4.41 ± 0.7

**Table 5 ijerph-16-04001-t005:** Removal efficiency of bioflocculant in comparison to chemical flocculants.

Type of Flocculants	Water Quality before Treatment			Water Quality after Treatment			Removal Efficiency (%)		
	BOD (mg/L)	COD (mg/L)	S (mg/L)	BOD (mg/L)	COD (mg/L)	S (mg/L)	BOD	COD	S
Bioflocculant	6.4 ± 0.0	1557 ± 0.0	4.1 ± 0.0	3.2 ± 0.2	436 ± 2.08	1.03 ± 0.13	59 ^b^	72 ^a^	75 ^a^
Alum	6.4 ± 0.0	1557 ± 0.0	4.1 ± 0.0	2.9 ± 0.2	828 ± 1	1.37 ± 0.12	50 ^a^	47 ^b^	66 ^b^
FeCl_3_	6.4 ± 0.0	1557 ± 0.0	4.1 ± 0.0	2.6 ± 0.58	753 ± 2.65	1.08 ± 0.12	54 ^a,b^	52 ^c^	73 ^a^

Different letters (a, b & c) denotes statistical significance at (*p* < 0.05).

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
