# Peer review of "Removal of Pollutants in Mine Wastewater by a Non-Cytotoxic Polymeric Bioflocculant from Alcaligenes faecalis HCB2"

_ijerph, 2019, doi:10.3390/ijerph16204001_

Round 1
Reviewer 1 Report
Specific comments
Abstract
Line 15, KCl should be changed to K+ Line 20, the authors should specify the cell line used for the toxicity assay.
Introduction
Line 40, should start a new paragraph
Materials and Methods
The control experiment for determination of bioflocculant was wrong; the author cannot you only kaolin suspension for the control with the cation and freshly prepared media or distilled water to replaced the cell-free supernatant. Line 94, Fe+3, Ca+2, Mn+2 and Ba+2 should be written as Fe3+, Ca2+, Mn+2 and Ba+2 Identification procedures of the bioflocculant-producing bacteria was not included in the method and the accession number given to the bacteria must be mentioned.
Results
Line 159, should be Results and discussion Line 173, “Agunbiadea et al [33]” should be written as “Agunbiade et al [33]” Line 175, “Brachybacterium sp,” should be written as “Brachybacterium” Line 177, “research work” should be replaced with “study” Line 178, “inoculum size” should be removed from the subheading Line 185, “negligible flocculating activity” should be replaced with “low flocculating activity” Line 199, Wu and Ye. [38] should be written as Wu and Ye [38] Figure 1. The interval unit of 50 for the shaking speed is not consistent. Double-check the x-axis again (165-220). Time course- flocculating activity and OD at time zero was not reported and these data are very important is this kind of assay. It is highly imperative to determine these values before fermentation to know the progress with increase in fermentation period. There is no significant different in the flocculating activity between the peak point and at 108 h and hence the authors didn’t supposed to stop the experiment at 108 h. Time course experiment should be stope when there is drastic decrease in flocculating activity. EDS results were not explained well. The potential sources of each of the element should be mentioned. The bioflocculant was said to be composed of carbohydrate and proteins, the sugar and the proteins monomeric units should be determined to suppose the proposed flocculation mechanism. Table 4. Zeta potential of different samples, where is the zeta potential of kaolin particles with Ba2+ coming from? Is Ba2+ the cation of choice for the effective flocculating activity of the bioflocculant? Application of bioflocculant on wastewater, procedures for COD and BOD should be properly stated in the method.

Author Response
All the comments have been addressed. The answers to each comments is written parallel to each question in a column.

Reviewer 2 Report
This manuscript is utilizing bioflocculants to work on the flocculation in the mine wastewater, which is a hot topic and meets the needs to solve the increasingly important environmental problems in the mining industry. However, overall it is not well organized while the data are pretty sufficient. Plus, the following concerns should be addressed before any further processes.
More information on the mine wastewater should be provided. In the Materials and Method section, some items are actually operating conditions, instead of the methods. Reorganize it if possible. In section 2.8, what the Fo means? It is the initial value of what? Optical density? Clarify it. As well as in other sections. In the measurement of Zeta potential, what is the concentration of the background KCl? And what is the pH values? The pH will cause the shift of surface potentials, even change the sign. What are the “a” and “b” meaning in Figure 1? The authors may want to more clearly describe the messages in tables and figures. In Figure 4, I understand the pH will change with time, but the initial pH may not be a good term. Delete “initial”. Why the scales(magnifications) in Figure 5 are not the same? The peaks in Figure 6 seem not enough, given so many elements in Figure 7. Why the x range in Figure 8 is 0-600°C, instead of 0-800/900°C? The weight is still decreasing at a temperature of 600 °C. The mechanism of the bioflocculant is not well-proposed. The results only show the biflocculant may behave better than conventional ones. The language needs to be polished carefully. Try to get rid of tense mistakes and clumsy phrases.Author Response
All the comments have been addressed. The answers to each comments is written parallel to each question in a column.

Reviewer 3 Report
The authors did a systematic study regarding the application of a bioflocculant in the treatment of kaolinite and metals. The test results are pretty encouraging and the mechanism was well explained. I would recommend acceptance after minor revision.
Line 77. Please explain why CaCl2 was added in the flocculation tests.
Line 82: People usually use turbidity to evaluate the flocculation performance. Turbidity of the supernatant will give a direct idea regarding the clarity of the solution. In this manuscript, what is the minimum turbidity of kaolinite flocculation supernatant (indicating the optimum test condition).
Line 197-201 and 93: In the optimization of culture conditions section, when the metal ions were added. Before or after the cell-medium separation? I thought the authors evaluated the effects of metal ions on the flocculation performance instead of bacteria incubation. However, the tests were listed in the Optimization of culture conditions section, which seems to be wired to me.
Line 229: What is the pH used for the flocculation tests?
Line 273: Basal and facial planes of kaolinite carry different charges.
Line 290: The SEM images quality is poor. It is difficult to distinguish the particles.
Line 327: EDX analysis should be performed on multiple areas since the inhomogeneity of the flocculant.
Line 337: I can see a weight change at 210 Celsius degree. However, I do not observe a significant change at the higher temperatures.
Line 355: I saw both ug/ul and mg/ml in the manuscript. Need to be revised for consistency.
Author Response

(The authors gave the same response as above.)

Round 2
Reviewer 2 Report
Overall the revised version has been improved. And most of my concerns are well addressed. But the mechanism section is still not sufficient and clear enough. Please refer to page 9-10 in paper "Selective Flocculation Separation of Fine Hematite from Quartz Using a Novel Grafted Copolymer Flocculant" by Zhang, et al. The possible interactions can be inferred from the Zeta potential results.
Author Response
Regarding the comments on the flocculation mechanism and the article you recommended to us. The authors in this article measured the zeta potential in different pH and we were expected to do the same. However, we measured the zeta potential at the pH in which the bioflocculant performed well (pH7) and this does not nullify our conclusion of the mechanisms involved in its flocculation activity. However, in the future we will consider the advice.